# Peptides that Mimic RS repeats modulate phase separation of SRSF1, revealing a reliance on combined stacking and electrostatic interactions

**Talia Fargason, Naiduwadura Ivon Upekala De Silva, Erin Powell, Zihan Zhang, Trenton Paul, Jamal Shariq, Steve Zaharias, Jun Zhang\***

Department of Chemistry, University of Alabama at Birmingham, Birmingham, United States

**Abstract** Phase separation plays crucial roles in both sustaining cellular function and perpetuating disease states. Despite extensive studies, our understanding of this process is hindered by low solubility of phase-separating proteins. One example of this is found in SR and SR-related proteins. These proteins are characterized by domains rich in arginine and serine (RS domains), which are essential to alternative splicing and in vivo phase separation. However, they are also responsible for a low solubility that has made these proteins difficult to study for decades. Here, we solubilize the founding member of the SR family, SRSF1, by introducing a peptide mimicking RS repeats as a co-solute. We find that this RS-mimic peptide forms interactions similar to those of the protein's RS domain. Both interact with a combination of surface-exposed aromatic residues and acidic residues on SRSF1's RNA Recognition Motifs (RRMs) through electrostatic and cation-pi interactions. Analysis of RRM domains from human SR proteins indicates that these sites are conserved across the protein family. In addition to opening an avenue to previously unavailable proteins, our work provides insight into how SR proteins phase separate and participate in nuclear speckles.

**\*For correspondence:**
zhanguab@uab.edu

**Competing interest:** The authors declare that no competing interests exist.

## Editor's evaluation

This study convincingly demonstrates that the splicing factor SRSF1 can be solubilized in the presence of short RS or ER containing peptides, and uses this discovery to determine the solution NMR structure of SRSF1, as well as to map its interactions with RS peptides. These findings are important in that SR proteins are key regulators of alternative splicing but their study has been greatly hampered by their low solubility. The development of a general method that allows their structural and biochemical analysis in solution will have broad applications.

## Introduction

Liquid-liquid phase separation underpins the formation of membraneless organelles, such as nucleoli (*Lafontaine et al., 2021*), P-bodies (*Brangwynne et al., 2009*), stress granules (*Molliex et al., 2015*), cajal bodies (*Neugebauer, 2017*), and nuclear speckles (*Fei et al., 2017*). The integrity of such organelles is maintained by interactions between biomolecules that form condensates, or liquid droplet-like structures, in which the local concentration of individual components is higher than the surrounding environment (*Yang et al., 2004*). These condensates cluster relevant molecules together to facilitate interactions while allowing rapid material exchange (*Yang et al., 2004*; *Souquere et al., 2009*; *Dundr and Misteli, 2010*; *Handwerger et al., 2005*). Mounting evidence has revealed roles of phase

separation in modulating reaction kinetics, enzyme catalysis, and binding specificity (*Reber et al., 2021*; *Banani et al., 2017*; *Strulson et al., 2012*; *Banjade and Rosen, 2014*; *Li et al., 2012*).

The protein SRSF1 (Serine/Arginine-Rich Splicing Factor 1, also known as ASF/SF2) is essential for the early-stage assembly of the spliceosome (*Kohtz et al., 1994*; *Cho et al., 2011*). Several in vivo studies have shown that SRSF1 is found in condensates (*Fei et al., 2017*; *Hammarskjold and Rekosh, 2017*; *Haward et al., 2021*; *Lamond and Spector, 2003*; *Azpurua et al., 2021*; *Ilik et al., 2020*; *Li and Wang, 2021*). Aberrant condensation behaviors have also been observed in disease states (*Azpurua et al., 2021*; *Ilik et al., 2020*; *Li and Wang, 2021*). SRSF1 belongs to the Ser/Arg-rich protein family (SR proteins), which contains 12 members possessing one to two structured RNA-recognition motifs (RRMs) and a repetitive Arg/Ser repeat region (RS domain; *Shepard and Hertel, 2009*; *Tacke and Manley, 1995*; *Screaton et al., 1995*). The RS region is also found in the much larger family of SR-related proteins, which contain the repetitive RS regions but not the other structural features (*Blencowe et al., 1999*; *Cascarina and Ross, 2022*). Two SR-related proteins, SRRM2 (serine-arginine rich repetitive matrix protein 2) and SON, are essential for the formation and structural maintenance of the membraneless organelles nuclear speckles (*Ahn et al., 2011*; *Xu et al., 2022*). Like many other splicing factors, SRSF1 modulates trafficking to the speckles (*Tripathi et al., 2012*). It has been demonstrated that SRSF1 is found in nuclear speckles when its RS domain is partially phosphorylated but that hyperphosphorylation causes the protein to leave nuclear speckles (*Aubol et al., 2018*; *Gui et al., 1994*). It is therefore evident that RS domains play an important role in the organization of nuclear speckles. However, an understanding of the nature of that interaction has been evasive due to a difficulty solubilizing the proteins involved. As with all 12 SR proteins and many speckle components, obtaining soluble SRSF1 has been an imposing challenge for decades, and this has substantially hindered our understanding of the functions of these proteins and of nuclear speckles as a whole (*Shepard and Hertel, 2009*).

Phase separation is frequently mediated by repetitive sequences. Here, we find this to be the case for the protein SRSF1, whose phase separation is dependent on its RS repeats. Our bioinformatic analysis reveals a correlation of RS repeats with a tendency to phase separate. We successfully solubilize SRSF1 using short peptides that mimic RS repeats. Our success in solubilizing SRSF1 provides an unprecedented opportunity to elucidate the mechanism by which RS repeats interact with SRSF1. We find that this increase in solubility is due to a competition between the peptide and RS domains for the same binding sites on RRM domains. We further discover that acidic residues and aromatic residues from SRSF1 RRMs interact with the RS region through salt bridges and cation-pi interactions. We find that many of the RRM sites interacting with RS repeats are conserved among the SR protein family. These findings provide insight into how interactions between SR and SR-related proteins may occur within membraneless organelles. They also allow us to predict how the nature of these interactions might change when the RS domain becomes phosphorylated.

## Results

### RS repeats are abundant in the human proteome and associated with phase separation

RS repeats are often found in SR proteins and SR-related proteins. The serine residues in RS repeats are frequently phosphorylated, a process which regulates the functions of RS repeats. To quantify the abundance of RS repeats in the human proteome, we systematically searched for uninterrupted repeats that were 2–8 amino acids in length (*Figure 1A*) and tested whether the length of RS repeats was correlated with protein condensates. Proteins were defined as being in condensates if they were listed in any of three available phase separation databases (PhaSepDB, LLPSDB, and DrLLPS) (*Ning et al., 2020*; *Li et al., 2020*; *You et al., 2020*). Because both RRM domains and RS repeats have been associated with phase separation in previous studies, we separated proteins containing RRM domains in this analysis from those that did not (*Wang et al., 2018*; *Murthy et al., 2019*). We found that the chance that a protein was found in condensates increased with the length of RS repeats it harbored regardless of whether the protein contained an RRM domain (*Figure 1A*). In our analysis, we did not distinguish RS repeats from SR repeats, and counting started with the first residue, whether it was R or S. We analyzed the correlation between the RS repeat length and the percentage of proteins found in condensates using correlation analysis (*Supplementary file 1*) and contingency

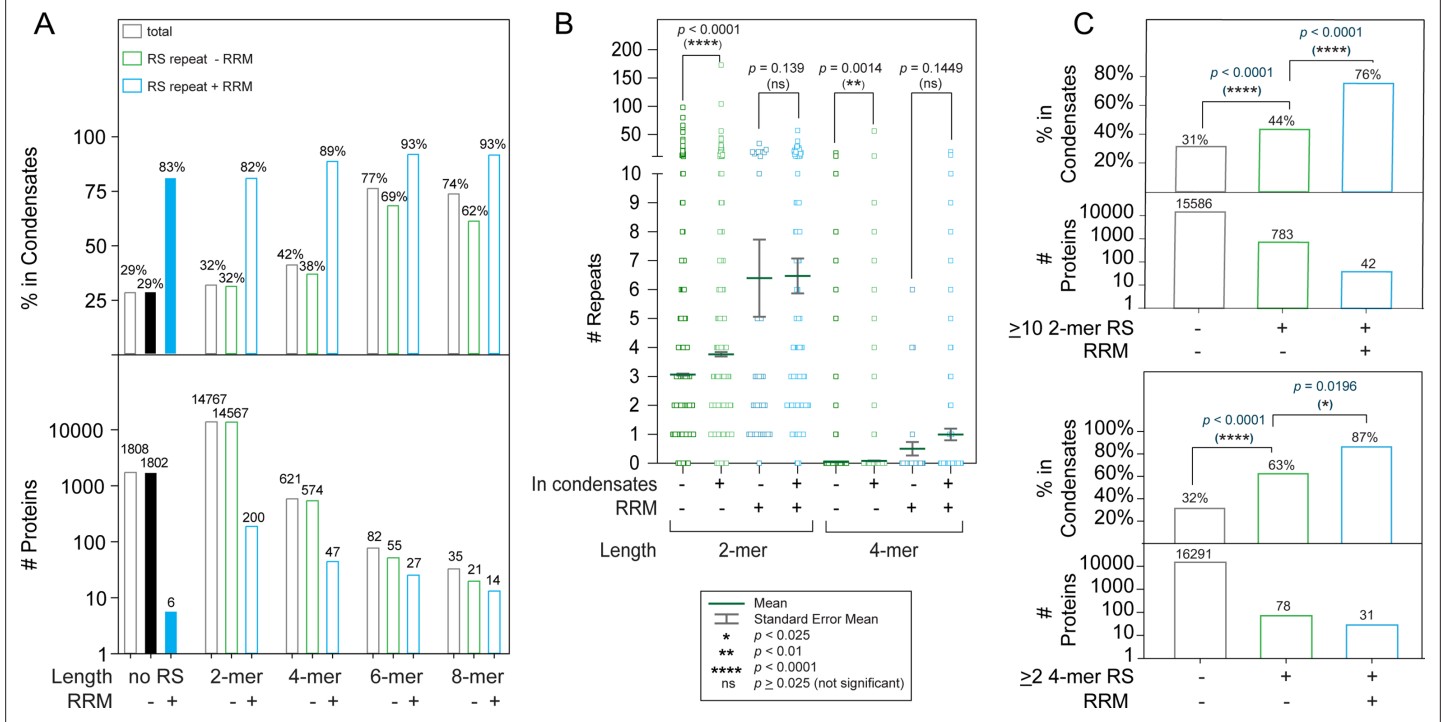

**Figure 1.** A combination of RS repeats and RRM domains is highly correlated with appearance in condensates. (**A**) Increased RS repeat length leads to an increased likelihood of appearance in condensates. Percentage of proteins possessing indicated properties that appear in one of three major phase separation databases. The Pearson's p-value (0.02) for the correlation between RS length and phase separation likelihood is shown in ***Supplementary file 1***. Correlation between RS and RRM occurrence was analyzed by Fisher's exact test (***Supplementary file 3***). (**B**) Correlation between number of 2-mer RS and 4-mer RS repeats with appearance in condensates. Proteins found in condensates are more likely to have a greater number of RS dipeptide and tetra-peptide repeats in the absence of RRM domains (-). The p-values presented were obtained using the Mann-Whitney test, which is suitable for non-normal distributions with different sample sizes (***Widen et al., 2020***). Bonferroni's adjustment was applied to adjust the significance level to p-value = 0.025. (**C**) Proteins with ≥10 2 mer RS repeats or ≥2 4 mer RS repeats are more likely to phase separate, particularly when RRM domains are present. p-values were calculated using Fisher's exact test (***Supplementary file 4***).

The online version of this article includes the following figure supplement(s) for figure 1:

**Figure supplement 1.** Increased repeat number and R/S percent composition correlate with increased phase separation.

tables (***Supplementary file 3***). Using correlation analysis, we found that the two-tailed Pearson's *p*-value is 0.02, and the correlation coefficient is 0.93 (***Supplementary file 1***). We further extended the correlation analysis to all other possible dipeptide motifs, assuming that the order of the two amino acids in a repeat is interchangeable. As the protein population of some dipeptide motifs is low, we estimated a population-based error of $1/\sqrt{N_{ps}}$, where $N_{ps}$ is the number of proteins containing 8-mer peptides found in condensates (as described more completely in the methods section). Applying a criterion of p-value <0.05 and fraction of proteins in condensates greater than twice the population-based error, we found six dipeptide motifs showed significant correlation with phase separation: GG, KK, QQ, PP, RG, and RS (***Supplementary file 1***). Except for the KK motif, the five other dipeptide repeats have been shown to directly drive phase separation for some proteins (***Lafontaine et al., 2021***; ***Brangwynne et al., 2009***; ***Molliex et al., 2015***; ***Fei et al., 2017***). It is noteworthy that among the six dipeptide motifs, RS-containing proteins have the highest percentage of 6-mer and 8-mer-containing proteins in condensates. As the sample sizes across these datasets vary widely, we performed the same analysis on 50 randomly selected size-matched subsets from each category (***Supplementary file 2***). We obtained similar results when size-matched datasets were used.

In addition to the effect of increased RS repeat length, we also found that proteins possessing both RS repeats and RRM domains were especially likely to be found in condensates (***Figure 1A***, blue bars). Among proteins with an RRM and at least one 4-mer RS repeat, the likelihood of appearance in condensates was 89%, and this trend became more pronounced as the repeat length was increased (***Figure 1A***). Using contingency tables, we analyzed the correlation between the presence

of RS dipeptides and the occurrence of RRM domains (*Supplementary file 3*). For proteins found in condensates, occurrence of RS dipeptides and RRM domains is clearly correlated, as shown by a p-value of 0.0013 (*Supplementary file 3*). In contrast, no significant correlation was found when the same analysis was performed on proteins not in condensates (p-value = 0.1202). As repeat length increased, the correlation between RS repeats and RRM domains also increased.

Many proteins contain multiple copies of short RS repeats. In fact, most SR and SR-related proteins have several short RS repeats instead of a few long, continuous ones (*Boucher et al., 2001*). One extreme example is nuclear speckle scaffolding protein SRRM2, which has 56 4-mer RS repeats, more than any other protein (*Figure 1—figure supplement 1A*). Therefore, we also analyzed how the number of short RS repeats is correlated with phase separation. We found that the number of RS repeats that a protein harbors also affects its likelihood of being found in condensates (*Figure 1B and C*). In the absence of an RRM domain, on average, proteins in condensates have more copies of 2-mer or 4-mer RS repeats than those not in condensates (*Figure 1B*). Further, proteins with several RS repeats and an RRM domain are particularly likely to be found in condensates (*Figure 1C*, *Supplementary file 4*). Definitions of RS domains usually specify either a threshold repeat number (*Boucher et al., 2001*; *Manley and Krainer, 2010*) or a threshold percentage R/S composition (*Cascarina and Ross, 2022*; *Manley and Krainer, 2010*). To test the effect of percent R/S composition on the likelihood of phase separation, we used LCD composer (*Cascarina et al., 2021*). Similar to the results we found for the effect of short repeats, we found that a 20-amino acid sequence of at least 40% RS composition increased the likelihood of a protein being found in condensates from 31% to 36% (*Figure 1—figure supplement 1B and C*, *Supplementary file 4*). Further addition of at least one RRM domain increased the likelihood of phase separation to 89% (*Figure 1—figure supplement 1C*). In summary, we found a correlation between RS repeats and phase separation whether it was analyzed by length, number, or composition of RS repeats.

## SRSF1 can be solubilized using peptides that mimic its RS region

The correlation of RS-repeats with phase separation is consistent with observations that many RS-containing proteins have low solubility in vitro. For example, up to this point, none of the full-length SR proteins have been obtained in concentrations suitable for biophysical/biochemical or structural characterization, although the founding member of the family, SRSF1, was identified more than three decades ago (*Krainer et al., 1990a*; *Ge and Manley, 1990*; *Krainer et al., 1990b*).

To overcome this obstacle in investigating SR and SR-related proteins, we aimed to develop a new purification and solubilization method using full-length SRSF1. The high Arg composition in these proteins inspired us to use high concentrations of Arg amino acid in our protocol to purify and solubilize SRSF1. An Arg/Glu mixture of 50 mM has been used to increase solubility of some RNA-binding proteins (*Golovanov et al., 2004*). We found that 0.8–1 M of Arg was able to solubilize all SRSF1 constructs during the purification procedure (details in the Materials and methods section). However, the high ionic strength of Arg at this concentration range is unsuitable for many analytical methods, such as NMR and binding assays.

We predicted that we could solubilize phase-separating proteins using peptide co-solutes that mimic these repeats to compete with inter- and intra- molecular interactions (*Figure 2A*). We tested this concept on SRSF1, which contains RS repeats of 16, 5, and 6 amino acids, respectively (*Figure 2B*). Serine residues in these regions can be phosphorylated, resulting in an alternation of positive and negative charges (*Figure 2B*). To solubilize SRSF1 in its unphosphorylated and phosphorylated forms, we therefore tested peptides of varying lengths to mimic unphosphorylated RS (RS), and phosphorylated RS (DR, and ER) repeats (*Figure 2B*). Here, using the purified protein, we found that SRSF1 phase separated at concentrations lower than 300 nM in a phosphate buffer (*Figure 2C and D*). This was also the case when the protein was diluted into 90 mM KCl (*Figure 2E*).

To quantify protein solubility, we used ammonium sulfate precipitation followed by resuspension of the proteins in peptide-containing buffers (*Figure 2F*). This approach to measuring protein solubility has been used in many studies (*Burgess, 2009*; *Trevino et al., 2008*). If mimicking the repetitive sequences with the peptides helps resolve phase separation, we expect to see a more dramatic solubility increase when the peptide co-solutes most closely mimic the repetitive sequences of the proteins. To this end, we measured solubility of unphosphorylated full-length SRSF1, hyper-phosphorylated SRSF1 (pi-SRSF1), and RS-truncated SRSF1 (ΔRS) (*Figure 2F*). We

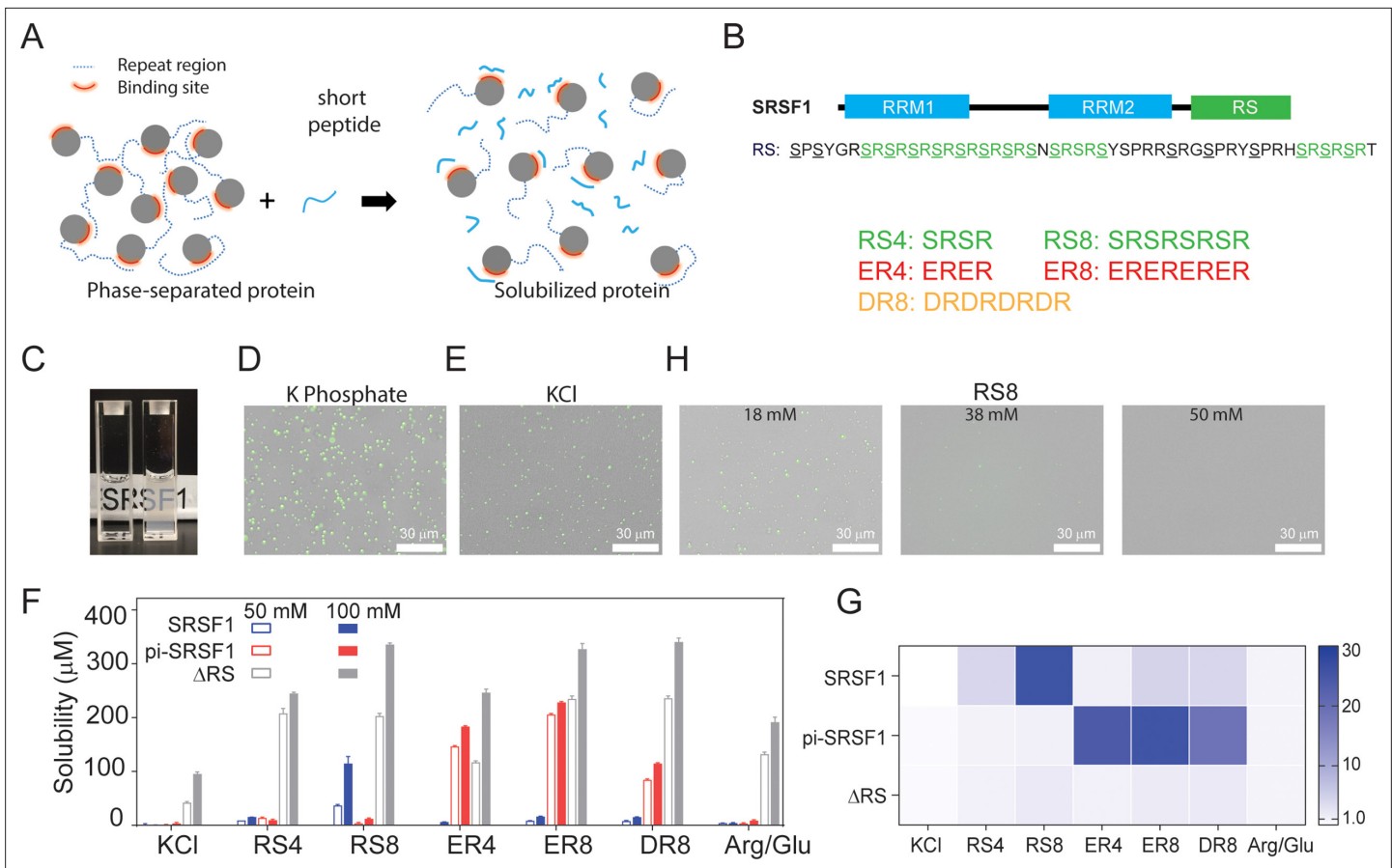

**Figure 2.** SRSF1 phase separation can be reduced using peptides that best mimic its RS repeats in their respective phosphorylation states. (**A**) Schematic illustration of solubilizing phase-separating proteins using short peptides. Short peptides compete with RS repetitive regions, disrupting phase separation. (**B**) Domain architecture of SRSF1. The underlined serine residues in the SRSF1 RS domain can be phosphorylated, and the phosphorylated RS can be mimicked by ER and DR repeats. Short peptide co-solutes used in this study are shown below. (**C**) Phase separation of SRSF1. The left cuvette is SRSF1 solubilized in the RS8 peptide, and the right cuvette is SRSF1 in 140 mM potassium phosphate, pH 7.4, 10 mM NaCl. The fluorescence image of 288 nM unphosphorylated SRSF1 in phosphate buffer (**D**), KCl buffer (**E**). SRSF1 is labeled with Alexa488 at N220C. (**F**) SRSF1 solubility using 50 mM or 100 mM of peptide as indicated. (**G**) Ratio of solubility in peptides to solubility in 100 mM Arg/Glu as determined in panel D. (**H**) The RS8 peptide can reduce phase-separation droplets of SRSF1.

The online version of this article includes the following figure supplement(s) for figure 2:

**Figure supplement 1.** Repetitive peptides solubilize other proteins containing similar repetitive regions.

found that whether phosphorylated or not, full-length SRSF1 was essentially insoluble in the 50 mM and 100 mM KCl control buffers (*Figure 2F*). Previous studies have found that truncation of the RS domain increases protein solubility (*Tacke and Manley, 1995*). This suggests that the RS domain is responsible for the protein's low solubility. To verify this, we measured the solubility of ΔRS and found that it was overall more soluble than full-length SRSF1 in all tested buffers (*Figure 2F*). Although an Arg/Glu mixture has been reported to promote protein solubility (*Golovanov et al., 2004*), Arg/Glu at 100 mM provided only a limited solubilizing effect for full-length SRSF1 (*Figure 2F*). Among the peptides we tested, only RS8 dramatically increased solubility of unphosphorylated SRSF1 (from 0.6±0.29 μM in 100 mM KCl to 120±12 μM with 100 mM RS8). Consistent with our hypothesis, all tested peptides had a less dramatic solubilizing effect on ΔRS, likely because it does not contain the repeat sequences that the peptides are designed to mimic. Hyper-phosphorylation of an RS domain converts the region into a basic acidic repeat resembling ER and DR repeats. To mimic a phosphorylated RS domain, we tested the solubilizing effects of ER and DR peptides (*Cho et al., 2011*; *Feng et al., 2012*). ER8 increased the solubility of hyper-phosphorylated SRSF (pi-SRSF1) more than other peptides. DR8 and ER4 also had substantial, albeit less pronounced, solubilizing effects (*Figure 2F*). The preference for ER8 may be due to the

fact that glutamic acid resembles phosphoserine more than aspartic acid in size. This confirmed our hypothesis that the solubilizing effect was more notable with peptides most closely resembling the proteins' own repeats.

These trends are clearer when the effect of the peptides on solubility is normalized by the solubility in 100 mM Arg/Glu (*Figure 2G*). Whereas peptides designed to mimic the repetitive constructs produce as much as a 30-fold increase in solubility relative to Arg/Glu, the solubility increase of ΔRS only reaches about a twofold difference (*Figure 2G*). In accordance with our solubility tests, we found that using increasing concentrations of the RS8 peptide reduced the number of liquid-like droplets in solution (*Figure 2H*).

We also tested the solubilizing effect of these peptide co-solutes on two other RNA-binding proteins, Nob1 and Nop9. These peptides increased solubility of Nob1 and Nop9 by 50–60% and 2–10%, respectively (*Figure 2—figure supplement 1A*). Nob1 contains unstructured regions rich in basic and acidic-basic residues (*Figure 2—figure supplement 1B*) and has an increased solubility despite its unstructured regions having a lower homology to the tested peptides. Nop9 does not have such unstructured sequence regions (*Figure 2—figure supplement 1C*). Consistent with our hypothesis, the tested peptides have a moderate solubilizing effect on Nob1, whereas they have limited or no effect on Nop9. These results for SRSF1 constructs, Nob1, and Nop9 suggest that repetitive peptides improve solubility for proteins that have similar sequences.

## Peptide co-solutes are compatible with NMR experiments and binding assays

Ionic co-solutes usually increase the dielectric constant of a sample and complicate NMR data acquisition, producing difficulty in probe tuning/matching, elongation of pulse width, and reduction of sensitivity (*Wider and Dreier, 2006*; *Kelly et al., 2002*). This imposes a considerable obstacle, as NMR is one of the few methods that provides an atomic level description of the dynamic interactions of phase-separating proteins. This adverse effect can be quantified by the elongation of the pulse width, which is inversely proportional to the signal sensitivity (*Wider and Dreier, 2006*). For example, increasing KCl concentration from 100 mM to 400 mM elongates the pulse width by about 50% on the NMR probe used in this study (*Figure 3A*). In contrast to the effect of KCl, peptide co-solutes did not significantly elongate the pulse width (*Figure 3A*). Whereas the 800 mM Arginine buffer used to solubilize SRSF1 during purification increased the pulse width to 16.97 μs, a combination of peptide and arginine had a less pronounced effect on the pulse width (*Figure 3—figure supplement 1A*). This is likely due to the low mobility of short peptides compared with salts or free amino acids (*Kelly et al., 2002*).

We expect the peptides to compete with homotypic inter-molecular interactions (interactions between SRSF1 molecules) to solubilize the protein, but the competition should not be strong enough to abolish binding or disrupt protein structure. With the peptide co-solutes, we were able to obtain high quality NMR spectra for both unphosphorylated and phosphorylated SRSF1. The TROSY-HSQC overlay in *Figure 3B* is consistent with the expected presence of both globular domains that show a higher level of dispersion and disordered regions with proton shifts in the 8.0–8.5 ppm range. Although we were able to solubilize unphosphorylated and phosphorylated SRSF1 in these respective buffers (RS8 and ER8), a buffer that solubilizes both proteins was desired to allow direct comparison of the two proteins and facilitate NMR assignment. Therefore, we examined the spectra and pulse width of different combinations of peptides and Arginine (*Figure 3—figure supplement 1*). We found that a buffer of 100 mM ER4 mixed with 400 mM Arg/Glu, pH 6.4, maintained structure and solubilized both unphosphorylated and phosphorylated SRSF1 constructs at concentrations above 350 μM (*Figure 3—figure supplement 1B*). It also weakened but did not abolish binding of SRSF1 constructs to an RNA ligand (*Figure 3—figure supplement 1C*) and resulted in a pulse width of 15.04 μs (*Figure 3—figure supplement 1A*), significantly shorter than that observed for 800 mM Arg/Glu. This buffer was used for future experiments. Using this buffer, we assigned the backbone for unphosphorylated SRSF1 (*Figure 3C and D*). This accomplishment enabled us to investigate the mechanism by which repetitive peptides solubilize SRSF1. To this end, we selected RS8 and unphosphorylated SRSF1 for further study.

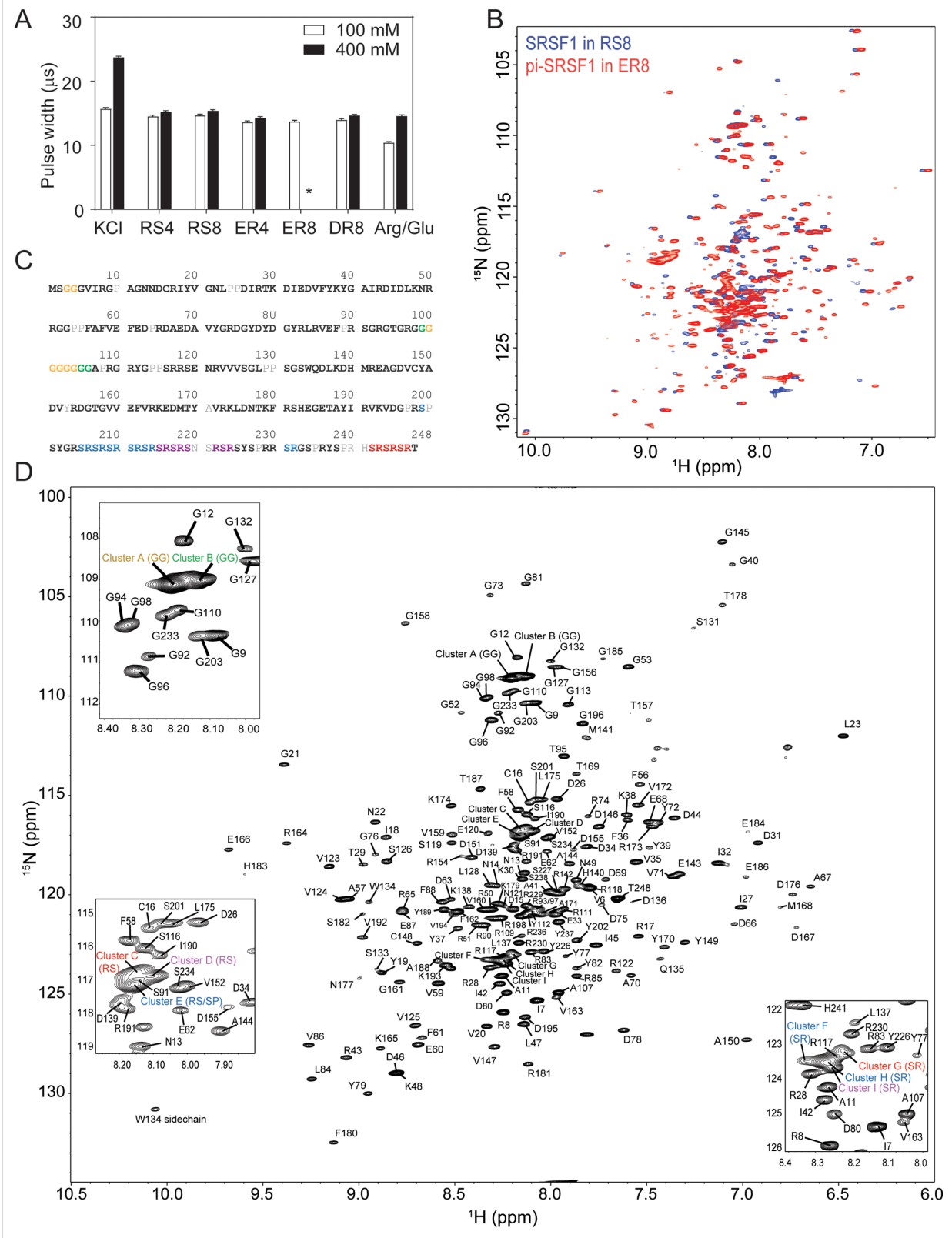

**Figure 3.** Short peptides are compatible with NMR experiments. (**A**) NMR 90 degree pulse width. ER8 is insoluble at 400 mM and therefore its pulse width could not be determined, as indicated by *. (**B**) [15]N-TROSY-HSQC overlay of SRSF1 in 100 mM RS8 and phosphorylated SRSF1 (pi-SRSF1) in 100 mM ER8. (**C**) Assigned residues in the SRSF1 protein sequence. Black bold fonts indicate non-overlapping residues. Gray fonts indicate unassigned residues. Color fonts indicate amino acids assigned to clusters. (**D**) Assignment of the SRSF1 amide groups.

*Figure 3 continued on next page*

*Figure 3 continued*

The online version of this article includes the following figure supplement(s) for figure 3:

**Figure supplement 1.** The buffer 100 mM ER4, 400 mM Arg/Glu, pH 6.4 was effective at solubilizing and optimizing spectral quality for both unphosphorylated and hyperphosphorylated SRSF1 at high enough concentrations for NMR assignment.

## Acidic and exposed aromatic residues of SRSF1 RRMs are responsible for the interactions with RS repeats that lead to phase separation

Mimic peptides were able to provide us with control over the critical point for SRSF1 phase separation, enabling us to obtain a backbone assignment in the solution state (*Figure 3*). According to our hypothesis, the mimic peptide should provide transient competition for contacts with the protein's repetitive sequence. Inter- and intra- molecular interactions should still occur under these conditions. However, they should be weakened enough to prevent droplet formation, providing us with a stable sample that can be used to study the intermolecular interactions that lead to the initiation of phase separation. To verify that this was the case, we performed a series of paramagnetic relaxation enhancement NMR experiments to probe peptide, intramolecular, and homotypic intermolecular interactions.

To locate the RS-mimic peptide interacting sites, we labeled RS8 with a paramagnetic probe (MTSL) and mixed it with SRSF1 (*Figure 4A and C*, and *Figure 4—figure supplement 1A*). The paramagnetic probe decreases intensities of residue peaks on the NMR spectrum in a distance-dependent manner (*Clore and Iwahara, 2009*). A higher PRE value indicates that peptides come closer to the residue analyzed. PRE is suitable for probing transient interactions, including the weak interactions between co-solutes and macromolecules (*Clore and Iwahara, 2009*; *Okuno et al., 2021*) and the intermolecular interactions that precede phase separation (*Murthy and Fawzi, 2020*; *Ryan et al., 2018*). We found that RS8 interacted primarily with RRM1 residues (*Figure 4A and C*, and *Figure 4—figure supplement 1A*). This is consistent with the fact that RRM1 (pI = 4.7) is more acidic than RRM2 (pI = 6.9), with regions of high negative charge on its two helices (*Figure 4B*). Dramatically perturbed sites were clustered on electronegative and aromatic sites, with the sequence $D^{31}IED$ on the $\alpha_1$-Helix and $D^{44}ID$ on the neighboring β-sheet being particularly perturbed (*Figure 4C*). Other hotspots included $D^{80}GYR$ and $E^{87}F$ on loops neighboring the $\alpha_1$ and $\alpha_2$ helices, respectively, $D^{66}AED$ on the $\alpha_2$ helix, and RRM2 residues W134 and $A^{150}DVYR$ (*Figure 4C*).

According to our hypothesis, the RS8 peptide should provide transient competition with the RS domain without abolishing inter- and intra-molecular interactions (*Figure 2A*). To locate the intramolecular interacting sites, we separately introduced the probe to the center of the RS domain (N220C, *Figure 4D*, *Figure 4—figure supplement 1B, D*) and the C-terminal end (T248C, *Figure 4—figure supplement 1C, E*). We found that labeling at the center of the RS domain produced the most notable perturbations (*Figure 4D*). To estimate the background PRE resulting from stochastic collisions, we also collected PRE data for SRSF1 mixed with the same concentration of probe alone as a control (*Figure 4—figure supplement 2A*). Subtracting the background PRE does not significantly change the perturbation pattern (*Figure 4—figure supplement 2D and E*). To verify that intermolecular interactions were not contributing to the measured intramolecular PRE, we performed a control PRE measurement by mixing an equal amount of probe-labeled ($^{14}$N, N220C-MTSL) SRSF1 and $^{15}$N-SRSF1 with no probe labeling (*Figure 4—figure supplement 2C*). Because $^{14}$N SRSF1 cannot be detected by NMR HSQC, in this experiment, observed PRE can only happen through intermolecular interactions. At this total protein concentration of 200 μM, we did not see a significant intermolecular contribution to the PRE signal. The relative strengths of the spectra are readily observed when the intermolecular interactions under these conditions are subtracted from the intramolecular interactions (*Figure 4—figure supplement 2F*).

To locate the inter-molecular interacting sites, we placed the probe on the C-terminal end of the protein and doubled the concentration to a total of 400 μM protein, maintaining a 1:1 ratio of HSQC-undetectable SRSF1 with the probe attached and $^{15}$N-labeled SRSF1 possessing no cysteines (*Figure 4E*, *Figure 4—figure supplement 1C and E*). With this experimental design, only intermolecular interactions resulted in PRE on the $^{15}$N-labeled SRSF1. Consistent with our hypothesis, the intra- and inter-molecular PRE patterns are similar to the perturbations from paramagnetic RS8.

To gain an atomic-level picture of the interactions between RS8 and SRSF1, we constructed models with the program Xplor-NIH using the PRE values as restraints (*Schwieters et al., 2006*;

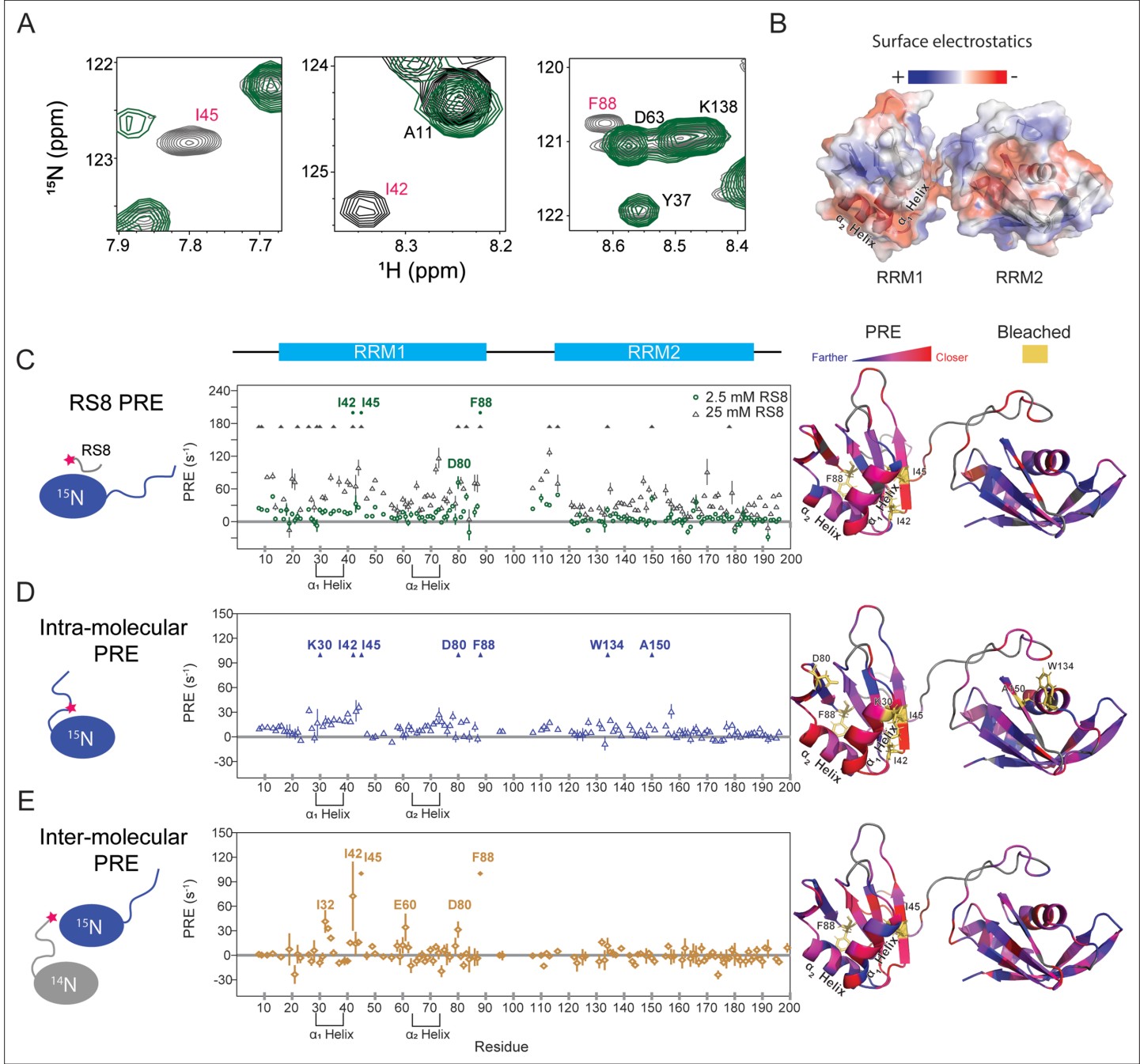

**Figure 4.** SRSF1 residues involved in interactions with the RS8 peptide are similar to those found in intra-, and homotypic inter-molecular interactions with the RS region. (**A**) 15N-TROSY-HSQC overlay of SRSF1 in 50 mM diamagnetic (gray) and 2.5 mM paramagnetic RS8 (green). The intensities of residues close to the probe become diminished. Bleached residues (indicated by red type) came in such close contact with RS8 that their intensities were diminished before the first observation time point (additional information in the methods section). The full spectra are shown in *Figure 4—figure supplement 1*. (**B**) Electrostatic surface of SRSF1 RRM1 and RRM2. The α1 helix on RRM1 has a large negatively charged surface area, and RRM1 possesses overall more negative charge. (**C**) PRE values induced by 2.5 or 25 mM paramagnetic RS8. (**D**) Intra-molecular PRE produced by the MTSL-labeled RS region (N220C). (**E**) Inter-molecular PRE produced by the MTSL-labeled NMR-inactive SRSF1 (T248C). The filled symbols indicate bleached residues. Yellow sticks in the molecular graphics on the right indicate bleached residues. Gray indicates residues whose PRE values are unavailable due to peak overlap or an inability to assign them. PyMOL molecular graphics were prepared using Xplor-NIH (see Materials and methods section for more information).

The online version of this article includes the following figure supplement(s) for figure 4:

*Figure 4 continued on next page*

*Figure 4 continued*

**Figure supplement 1.** SRSF1 residues involved in interactions with the RS8 peptide are similar to those found in intra-, and homotypic inter-molecular interactions with the RS region.

**Figure supplement 2.** SRSF1 residues involved in interactions with the RS8 peptide are similar to those found in intra-, and homotypic inter-molecular interactions with the RS region.

*Schwieters et al., 2003*) as illustrated in *Figure 5—figure supplement 1*. We further optimized these models using molecular dynamics simulations (*Figure 5*, *Figure 5—figure supplement 2*). The top 25% of initial Xplor-NIH structures agreed with the observed PRE data, with Pearson's correlation coefficients of 0.916–0.941 (*Figure 5—figure supplement 1*). MD simulations produced structures in which a paramagnetic center was within the expected 12–15 Å from bleached residues (*Supplementary file 5*; *Iwahara et al., 2007*). Representative images are displayed in *Figure 5*. The hotspot $D^{31}IED$ on the $\alpha_1$ helix was found to be able to provide electrostatic contacts for multiple interactions, including hydrogen bonding with bleached isoleucine residues I42 and I45 (*Figure 5A*). In the region surrounding W134, an electrostatic interaction with D151 was found to enable a peptide arginine to orient parallel to the aromatic face of W134, forming cation-pi stacking interactions (*Figure 5B*). Simultaneous cation-pi stacking interactions were also observed in the regions surrounding D80 and Y79 (*Figure 5C*) as well as F88 (*Figure 5D*).

## SRSF1 RRM sites involved in phase separation are conserved in the SR protein family

We were curious whether interactions found in SRSF1 were conserved across the SR protein family. To this end, we used the program ClustalX (*Larkin et al., 2007*) to align the RRM1 sequences of SR proteins and found that interaction hotspots on α1 and α2 helices of RRM1 domains were conserved in most SR proteins (*Figure 6*). The electronegative charge was most highly conserved in the α2 helix and the neighboring loop, while in the α1 helix, acidic residues were replaced by arginine residues in some SR proteins. RRM1 domains have a conserved RNA binding site (*Figure 6*, sticks). Interestingly, these regions are distal to the RNA-binding sites of the RRM domains, suggesting the conservation

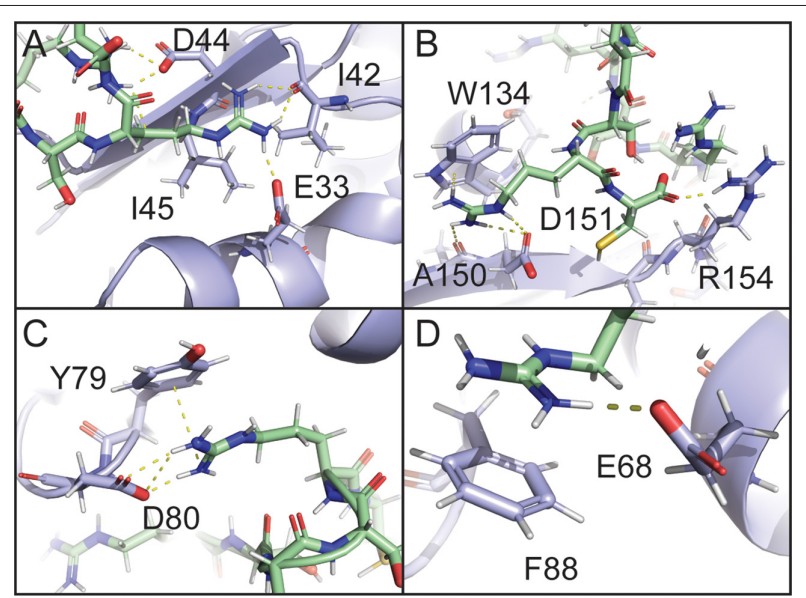

**Figure 5.** Electrostatic and cation-pi interactions are responsible for intermolecular interactions. Molecular dynamics simulation of SRSF1 with four RS8 peptides.

The online version of this article includes the following figure supplement(s) for figure 5:

**Figure supplement 1.** Fitting of MD simulation structure to PRE values.

**Figure supplement 2.** MD simulation trajectories of SRSF1 ΔRS interacting with four RS8 peptides.

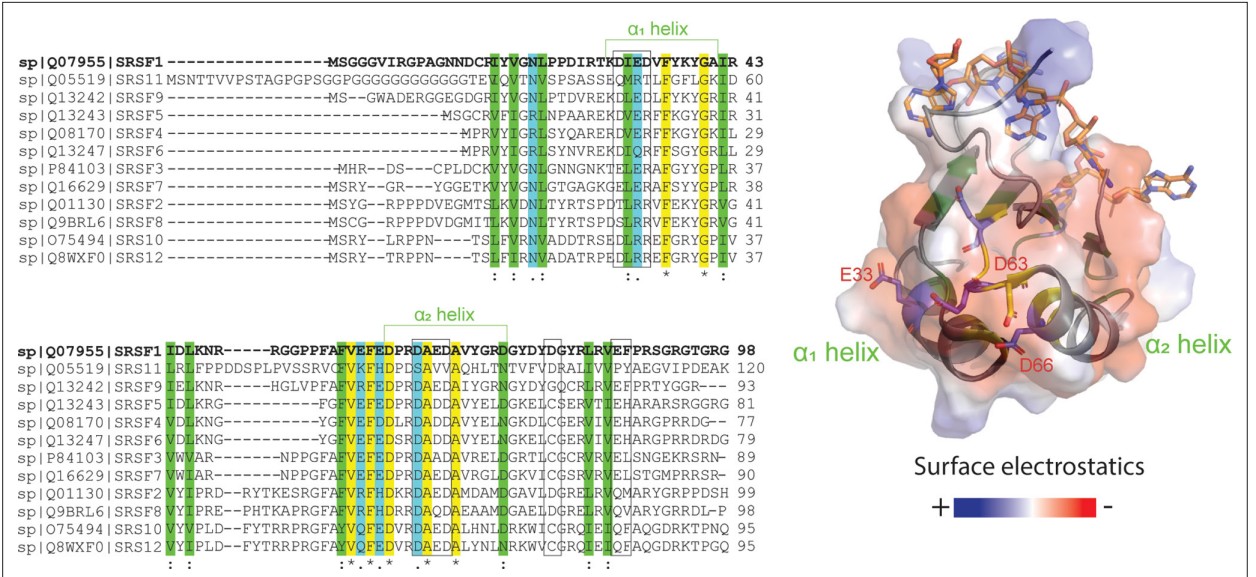

**Figure 6.** RRM1 residues responsible for SRSF1 phase separation are conserved throughout the SR protein family. ClustalX alignment of RRM1 domains of the SR protein family, where yellow indicates identical amino acids, green and blue indicate conserved residues. Black boxes indicate PRE hotspots. Structure of RNA-bound SRSF1 RRM1 was obtained from PDB ID 6HPJ. Transparent electrostatic surface is displayed. Conserved electronegative residues opposite the RNA binding pocket are shown in sticks.

may be due to a role other than RNA recognition (*Figure 6*). Considering the conservation of these sites involved in phase separation, the phase-separating mechanism we revealed for SRSF1 could be applicable to many other members of the SR family.

## Discussion

There is currently a need to improve methods for determining which proteins phase separate and by what mechanism they do so, but solubility concerns make isolated experiments out of reach for many proteins. To solubilize phase-separating proteins, denaturants or high concentrations of salts are typically used. Denaturants are unsuitable for experiments characterizing native state proteins. High concentrations of salts are flawed as they interfere with NMR (*Wider and Dreier, 2006*), SAXS (*Putnam et al., 2007*), and circular dichroism (*Greenfield, 2006*). Some proteins experience salting out when ionic co-solutes are introduced (*Murthy et al., 2019*; *Ryan et al., 2018*; *Martin et al., 2021*). For example, we found that the solubility of SRSF1 is around 2–7 µM in 1–5 M of NaCl (data not shown).

The protein solubilizing strategy used here is of wide applicability, not just confined to the examples of SRSF1 and Nob1. Our bioinformatic search revealed that RS-containing proteins are highly abundant and that there is a positive correlation between these repeats and phase separation. In addition to RS repeats, GG, KK, QQ, PP, and RG demonstrated both a strong positive correlation between repeat length and phase separation and robust enough sample sizes to render these results significant (*Supplementary file 1*). However, it is important to note that the repeats mentioned above are likely not a comprehensive list of repetitive sequences conducive to phase separation. In fact, aside from LL, SG, PL, and TA repeats, all dipeptides that exist in 8-mer sequences possess at least a weak positive correlation between repeat length and tendency to appear in condensates (*Supplementary file 1*). It is also important to note that with our current knowledge, there is a possibility for both overestimation and underestimation of phase-separating proteins when using these databases. Some proteins that phase separate may not yet be identified. Further, proteins reported to be in condensates do not necessarily phase separate on their own. It is also possible that a protein can have more than one type of repeated motif mediating phase separation. For these reasons, in vitro phase separation experiments using purified protein are imperative. We hope that the use of this method will expand the number of techniques available to perform such experiments. Development of peptide

structure-activity relationships (SARs) may serve as a technique for identifying the repeats responsible for phase separation as well. For instance, if a SAR reveals that a protein with more than one repetitive sequence reaches optimum solubility with one particular peptide mimic, the repetitive sequence corresponding to the peptide mimic may contribute more towards driving phase separation. Likewise, if a SAR reveals that a mixture of multiple mimic peptides is optimal for enhancing solubility, it is possible that multiple repetitive sequences within the protein are responsible for phase separation.

Characterization of intermolecular interactions that occur in the dispersed (soluble) state is an accepted method of understanding what interactions lead to phase separation (*Murthy and Fawzi, 2020*; *Ryan et al., 2018*; *Emmanouilidis et al., 2021*). A previous comparison of intermolecular PRE spectra of the protein FUS in the dispersed versus the condensed (phase separated) state indicated that the transient intermolecular interactions seen between molecules in the solution state are comparable to the intermolecular contacts seen when the protein is phase separated (*Murthy et al., 2019*; *Monahan et al., 2017* as discussed in *Murthy and Fawzi, 2020*). Our technique is unique in that it provides transient competition for the intermolecular interactions that lead to phase separation without abolishing these interactions entirely. We find that both RNA binding (*Figure 3—figure supplement 1*) and homotypic intermolecular interactions (*Figure 4E*) can still occur in a peptide-containing buffer. However, the presence of the peptide weakens intermolecular interactions enough to allow high quality NMR spectra to be obtained.

Here, we find that targeted competition for intermolecular interactions provides direct control over the critical point for phase separation, enabling experiments to be performed in the dispersed state that otherwise might not be possible. The ability to compare dispersed and condensed states for more proteins is still desirable. NMR spectra of isolated low complexity domains in the condensed state have been obtained successfully for several proteins including HNRNPA2, FUS, and a Caprin1-pFMRP complex (*Murthy et al., 2019*; *Ryan et al., 2018*; *Wong et al., 2020*; *Kim et al., 2019*; *Burke et al., 2015*). However, interactions between the structured domains and low complexity domains of these proteins have not yet been probed using these techniques. One bottleneck to obtaining usable condensed state samples involves obtaining high concentrations of soluble protein before inducing phase separation in the sample (*Murthy et al., 2019*; *Ryan et al., 2018*; *Wong et al., 2020*). Due to the high concentrations needed for an NMR backbone assignment and the negative effect of sample viscosity on NMR spectral quality, it is also more practical to perform backbone assignments of proteins in the dispersed states (*Murthy et al., 2019*; *Ryan et al., 2018*; *Wong et al., 2020*). We hope our method may serve as a useful tool in expanding these techniques.

Homotypic intermolecular interactions and interactions with the peptides involve residues similar to those involved in intramolecular interactions. However, intermolecular interactions seem to have a greater preference for the more negatively charged RRM1. Whereas interactions with RRM1 appear at all concentrations studied, a concentration of 25 mM peptide is needed to observe bleaching of W134 and A150 on the electropositive RRM2 domain (*Figure 4C*). This difference may be due to the fact that, while intramolecular interactions involve an RS domain tethered to RRM2 that helps facilitate interaction, external RS repeats do not have a method of compensating for this charge repulsion.

This preference for RRM1 is interesting because the interactions seen on RRM2 involve the same residues that bind to RNA (*Cléry et al., 2013*), but the interactions on RRM1 are opposite the RNA-binding interface (*Cléry et al., 2021*). In fact, chemical shift perturbations performed in a previous study indicate that the α1 helical residues on RRM1, in particular, remain virtually unaffected when two different RNA ligands are introduced, indicating there are also no allosteric effects (*Cléry et al., 2021*). Further, negative charges in this region are conserved across multiple SR protein family members (*Figure 6*). This suggests that RRM domains of SR proteins may have alternative sites used to mediate the protein-protein interactions that lead to phase separation. This is important because it means that the effect of phase separation on RNA binding can potentially be studied by disruption of these distal sites.

As we learn more about biomolecular condensates, it is of interest to understand what causes proteins to migrate to one condensate over another. It has been shown that the isolated SRSF2 RRM can localize to speckles on its own (*Greig et al., 2020*), which suggests there may be an additional molecular grammar within the structured components of these proteins that directs them towards nuclear speckles. It is known that speckles rely on RS repeats as scaffolds, as truncation of SRRM2's regions containing RS repeats (*Figure 1—figure supplement 1*) disrupts speckles (*Ilik et al., 2020*;

*Xu et al., 2022*). It is possible that these RS repeats function in part by providing multiple interaction sites for this type of RRM.

## Ideas and speculation

We demonstrate that electronegative $\alpha_1$ and $\alpha_2$ helices along with neighboring aromatic residues serve as interacting sites for unphosphorylated RS repetitive sequences. This finding has implications for how phosphorylation might change interactions within the speckles. If negatively charged residues are important for maintaining protein-protein interactions, charge repulsion between a hyperphosphorylated tail and acidic residues might be one reason that SR proteins leave the speckles upon hyperphosphorylation (*Gui et al., 1994*). Phosphoserines of RS repeats have been proposed to form salt bridges with neighboring arginine residues (*Hamelberg et al., 2007*), although any such contacts are likely short-lived and do not result in a stable secondary structure (*Ngo et al., 2008*; *Xiang et al., 2013*). Temporary phosphoserine-arginine contacts may be sufficient to compete with the highly transient cation-pi stacking interactions that we observe here.

# Materials and methods

## SRSF1 expression and purification

The DNA encoding human SRSF1 was sub-cloned into pSMT3 using BamH I and Hind III. The ΔRS construct and mutants SRSF1 C16S C148S N220C (N220C), SRSF1 C16S C148S T248C (T248C), and SRSF1 C16S C148S (NoC) were prepared using mutagenesis PCR. All these mutants maintain the folded structure according to NMR spectra, and bind with SRSF1 cognate RNA UCAGAGGA. All proteins were expressed by BL21-CodonPlus (DE3) cells in LB media or minimal media supplemented with proper isotopes for NMR experiments. Hyperphosphorylated SRSF1 was prepared by co-transformation of BL21-CodonPlus (DE3) cells using pSMT3/SRSF1 and CDC2-like kinase 1 (CLK1) cloned in pETDuet-1. Cells were cultured at 37 °C to reach an OD600 of 0.6, and 0.5 mM IPTG was added to induce protein expression. Cells were further cultured 16 hours at 22 °C. The cells were harvested by centrifugation (4000 RCF, 15 min). The cell pellet was re-suspended in 20 mM HEPES, pH 7.5, 150 mM Arg/Glu, 2 M NaCl, 25 mM imidazole, 0.2 mM TCEP supplemented with 1 mM PMSF, 1 mg/mL lysozyme, 1 tablet of Pierce protease inhibitor, and 1 mM NaVO4 for the hyperphosphorylated construct. After three freeze-thaw cycles, the sample was sonicated and centrifuged at 23,710 g for 40 min using a Beckman Coulter Avanti JXN26/JA20 centrifuge. The supernatant was loaded onto 5 mL of HisPur Nickel-NTA resin and then eluted with 60 mL of 20 mM MES pH 6.5, 300 mM imidazole, 600 mM Arg/Glu, and 0.2 mM TCEP. The eluted sample was cleaved with 2 µg/mL Ulp1 for 2 hr at 37 °C. The four unphosphorylated SRSF1 constructs (WT, N220C, T248C, NoC) were further purified by a 5 mL HiTrap Heparin column. The hyperphosphorylated SRSF1 was further purified by a 5 mL Cytiva Fast Flow Q column. The eluted samples from the ion exchange step were further purified by a HiLoad 16/60 Superdex 75 pg size exclusion column equilibrated with 800 mM Arg/Glu, pH 6.5, 1 mM TCEP, 0.02% NaN₃. The protein identities were confirmed by mass spectrometry. As reported in previous study, 18-phosphates were added on the RS region of SRSF1 (*Aubol et al., 2013*). The protein purities were judged to be >95% based on SDS-PAGE.

Between purification and NMR experiments, the protein was transferred to peptide buffer in one of three ways: (1) It was concentrated in 800 mM Arg/Glu pH 6.5, 1 mM TCEP, 0.02% NaN₃ and diluted with a peptide buffer to the final concentration (For *Figures 3D and 4D–E*, *Figure 3—figure supplement 1*, *Figure 4—figure supplement 1*, *Figure 4—figure supplement 2*). (2) It was precipitated and re-suspended in the peptide buffer (for *Figure 3B*). (3) For the peptide titrations (*Figure 4C*), because a low concentration was needed, the initial spectrum was taken in 200 mM Arg/Glu, and peptide was titrated into the NMR tube.

## Nop9 and Nob1 expression and purification

Nop9 and Nob1 expression and purification are detailed in published papers (*Zhang et al., 2016*; *Lamanna and Karbstein, 2009*). SUMO-tagged proteins were induced by 0.4 mM IPTG and expressed at 22 °C overnight in *E. coli* strain BL21-CodonPlus (DE3). The LB miller medium was supplemented with 0.1 mM ZnSO₄ for Nob1 expression. Cell pellets were re-suspended in 25 mM HEPES, pH 7.5, 1 M NaCl, 1 mM TCEP, 25 mM imidazole, 1 mg/mL lysozyme and lysed by sonication, followed by

centrifugation. The supernatant was applied to HisPur Ni-NTA resin, washed with 200 mL of loading buffer, and eluted with 25 mM HEPES, pH 7.5, 500 mM NaCl, 1 mM TCEP, 500 mM imidazole. The SUMO tag was cleaved overnight with 2 µg/mL of Ulp1 at 4 °C. The cleaved sample was purified by a 5 mL HiTrap Heparin column (GE Healthcare), and polished using a HiLoad 16/60 Superdex 200 column (GE Healthcare) equilibrated in 25 mM HEPES, pH 7.5, 500 mM NaCl, and 1 mM TCEP. The protein purities were >95% based on SDS-PAGE.

## NMR assignment

SRSF1 cultured in $^{2}$H,$^{13}$C, $^{15}$N M9 media was concentrated to 370 µM in 100 mM ER4, 400 mM Arg/Glu, pH 6.4, 1 mM TCEP, 10% D$_2$O, and 0.02% NaN$_3$. Triple resonance assignment experiments HNCA, HNCACB, HN(CO)CA, HN(CO)CACB, HNCO, and HN(CA)CO were collected at 37 °C on a Bruker Avance III-HD 850 MHz spectrometer installed with a cryo-probe. Approximately 85% of the protein backbone region was assigned using this method. Another approximately 13% of backbone exists in the disordered state with highly degenerate sequences, which leads to heavy peak overlap. These RS and G-rich regions were grouped into clusters. Multiplicity selective in-phase coherence transfer (MUSIC) experiments were collected to further characterize the clusters and verify the assignment of the rest of the protein. MUSIC was performed on SRSF1 for the following amino acids: Ser, Arg, Thr, Asn, Ala, Tyr/His/Phe, Pro, Asn/Gln, Met, and Gly. When used in combination with analysis of the effect of paramagnetic tag placement, peak clusters were able to be assigned to locations on the disordered regions. The NMR data was processed using NMRPipe (*Delaglio et al., 1995*), and assignment was performed using NMRViewJ (*Johnson, 2004*). The assignment of the well dispersed regions (85% of the protein) has been submitted to BMRB (ID: 51299).

## Paramagnetic relaxation enhancement (PRE) measurements

RS peptide with a sequence 'SRSRSRSRC' was synthesized and purified by GenScript with a purity >98%. The cysteine residue at the C-terminus was introduced for MTSL labeling. RS peptide was mixed with MTSL in a molar concentration ratio of 1:4. The pH was adjusted to 7.0 before a 2 hr labeling at room temperature. To remove unreacted MTSL, 10 mL of ether was added to the sample, and the mixture was vortexed and spun at 4000 rpm for 5 min. The extraction process was repeated twice. After purification, the pH of the peptide was adjusted to 6.5 using KOH, and the peptide was lyophilized. $^{1}$H paramagnetic relaxation enhancement (PRE) data was gathered at 37 °C on a Bruker Avance III-HD 850 MHz spectrometer installed with a cryo-probe. Titrations were performed by adding solid peptide to a $^{15}$N-labeled SRSF1 construct without cysteine (the NoC construct) in 200 mM Arg/Glu, pH 6.3, 1 mM TCEP. PRE spectra were obtained for the protein in 200 mM Arg/Glu alone, 200 mM Arg/Glu with 2.5 mM peptide, 200 mM Arg/Glu with 25 mM peptide, and 200 mM Arg/Glu with 50 mM peptide. After the spectrum with 50 mM peptide was collected, MTSL was quenched using 10 mM sodium ascorbate, and a PRE experiment was run on the quenched sample.

To prepare the sample for inter-molecular PRE, an SRSF1 construct with one cysteine at the C-terminal end of the RS tail (T248C) was obtained using mutagenesis PCR, and the mutated protein was expressed by BL21-CodonPlus (DE3) cells in LB media. The protein was exchanged into an MTSL labeling buffer of 0.8 M Arg/Glu, 100 mM NaCl, 50 mM Tris-HCl pH 7.0 using a HiPrep 26/10 desalting column. The sample was diluted to a concentration of 20 µM, and MTSL was added to a concentration of 0.4 mM. The sample was incubated in the dark at 37 °C for 12 hr, after which a desalting column was used to remove unreacted MTSL. The MTSL-labeled, NMR-inactive SRSF1 was mixed in a 1:1 ratio with a $^{15}$N SRSF1 construct with no cysteine (NoC) in a buffer of 100 mM ER4, 5 mM MES, pH 6.4, 400 mM Arg/Glu, and 5% D2O. The final concentration of protein was 420 µM (210 µM SRSF1 C16/148 S T248C-MTSL and 210 µM $^{15}$N SRSF1 C16/148 S).

To measure intramolecular PRE, an SRSF1 construct with one cysteine at the center of the first RS domain (SRSF1 C16S/C148S/N220C) was obtained using mutagenesis PCR and the purification method described above with the growth performed in M9 media containing $^{15}$N isotopes. MTSL was labeling was performed as described above. The final concentration of protein was 220 µM.

The low concentration intermolecular PRE was collected between SRSF1 C16/148 S N220C-MTSL and $^{15}$N SRSF1 C16/148 S at a total concentration of 185 µM (93 µM of each construct) as described above. A control PRE experiment with free MTSL was conducted by adding 220 µM MTSL to 220 µM $^{15}$N SRSF1 C16/C148S (NoC) in the NMR buffer described above.

All PRE measurements were carried out using a pulse sequence developed by Junji Iwahara (*Iwahara et al., 2007*). Diamagnetic data were collected after adding 2 mM ascorbic acid. The NMR data was processed using NMRPipe (*Delaglio et al., 1995*) and analyzed using NMRViewJ (*Johnson, 2004*). PRE values and errors were estimated as described previously (*Iwahara et al., 2007*).

Residues were considered above noise level if their intensities at the second time point on the diamagnetic spectrum were greater than five times the standard deviation of the spectrum. Peaks below this noise level threshold were excluded from analysis on all spectra. Peaks in regions of high spectral overlap were also excluded. If at the first time point of collection (approximately 12ms after the first 90° pulse), the intensity of the paramagnetic peak was less than or equal to half of the intensity of the diamagnetic peak at that time point, the peaks were was defined as bleached. In these circumstances, the relaxation occurred too quickly to allow fitting of the exponential decay curve. Residues G52 and R154 met the definition of both bleached and noisy. They were excluded from analysis.

## Solubility assays

Purified protein aliquots (40 μL) were incubated with 3.2 M ammonium sulfate on ice for 30 min before 10-min centrifugation at 14,000 RCF at 4 °C. After confirming that no protein was present in the supernatant, the supernatant was discarded. The pellets were re-suspended in 20 μL of corresponding buffers and shaken at room temperature for 30 min. The re-suspensions were further centrifuged at room temperature at 14,000 RCF for 5 min. The protein concentrations in supernatants were measured using UV absorbance at 280 nm. Error bars represent standard deviation from three technical repeats. The initial concentration of full-length SRSF1 constructs was 250 μM. As RS-deleted SRSF1 has a higher solubility, the initial protein concentration used for this construct was 400 μM.

## Molecular graphics

An Alphafold structure was downloaded from the Uniprot website for SRSF1 ΔRS and refined using Xplor-NIH. Restraints used for refinement included dihedral angles obtained from the assignment, RDC values obtained in a previous study (*Fargason et al., 2020*), and chemical shift perturbations. PRE values were projected onto to the structure in PyMOL by reassigning B-factors and coloring on a ramp scale.

## MD simulations

An Alphafold structure was downloaded from the Uniprot website for SRSF1 ΔRS and refined using Xplor-NIH. Docking of peptides was accomplished with Xplor-NIH using a restrained rigid-body simulated annealing protocol refined against the PRE, CSP, RDC, and dihedral angle data. In total, 100 Xplor-NIH structures were calculated using an ensemble size of 10. Each ensemble member had a single peptide, resulting in 10 total peptides in the model. The RRM1 domain (residues 16–90) was held in place while the N-terminus and linker were allowed full flexibility. RRM2 residues (residues 121–196) were allowed to move as a group. Linker and N-terminal residues were allowed full flexibility.

The general distance relationship for PRE is defined as *Iwahara et al., 2007*:

$$\Gamma_2 = \frac{1}{r^6}\left(\frac{\mu_0}{4\pi}\right)^2 \frac{1}{15}\gamma_I^2 g^2 \mu_B^2 S(S+1)\left(4\tau_c + \frac{3\tau_c}{1+(\omega_H\tau_c)^2}\right) \tag{1}$$

where $\Gamma_2$ is the PRE value, r is the distance between the paramagnetic center and the observed nucleus, $\mu_0$ is the vacuum permeability constant, $\gamma_I$ is the nuclear gyromacnetic ratio, g is the electron g-factor, $\mu_B$ is the electron Bohr magneton, S is the electron spin quantum number, $\omega_H/2\pi$ is the nuclear Larmor frequency, and $\tau_c$ is the PRE correlation time (where $\tau_c^{-1} = \tau_r^{-1} + \tau_s^{-1}$, $\tau_r$ = nuclear rotational correlation time, $\tau_s$ = electron relaxation time).

Because of the flexible nature of the protein, the structures were described using multiple ensemble members. For each residue (*h*), the PRE was determined by the average distance across the ensemble members:

$$\left\langle r^{-6}\right\rangle_h = \frac{1}{N}\sum_i^N r_i^{-6} \tag{2}$$

where N is the number of ensemble members, and $r_i$ is the distance between the paramagnetic center (the nitroxy oxygen of MTSL) and the nucleus under observation (the amide proton) in a single ensemble. Angle brackets (<>) indicate ensemble averages. The PRE values were back-calculated using the SBMF mode described in *Iwahara et al., 2004*:

Agreement between experimentally observed PRE and back-calculated PRE was assessed using the Q-factor (Q) and Pearson Correlation coefficients (R):

$$Q = \sqrt{\frac{\sum_{h}^{n} \left\{ \Gamma_2^{obs}(h) - \Gamma_2^{calc}(h) \right\}^2}{\sum_{h}^{n} \Gamma_2^{obs}(h)^2}} \tag{3}$$

$$R = \frac{\sum_{h}^{n} (\Gamma_h^{obs} - \overline{\Gamma^{obs}})(\Gamma_h^{calc} - \overline{\Gamma^{calc}})}{\sqrt{\sum_{h}^{n} (\Gamma_h^{obs} - \overline{\Gamma^{obs}})^2} \sqrt{\sum_{h}^{n} (\Gamma_h^{calc} - \overline{\Gamma^{calc}})^2}} \tag{4}$$

where n is the number of residues for which PRE values were obtained.

The top 25% of structures possessed Pearson correlation coefficients between 0.916 and 0.941 and Q-factors between 0.454 and 0.548.

These Xplor-NIH structures were used to produce an MD-simulation starting structure with 4 peptides and 1 SRSF1 ΔRS structure. The structure was further refined using AMBER20 with the ff19SB forcefield. Solvation was performed with explicit TIP3P water molecules with 0.15 M NaCl used to balance the charges. The simulation temperature was set to 300 K, and the cutoff distance of nonbonded interactions was set to 10 Å. A simulation in which no restraints were applied was run for 201 ns. This simulation accounted for bleached residues, which, with the exception of residue 88, remained within the expected 12–15 Å from the paramagnetic probe. For residue 88, a separate simulation was run for 17 ns in which a distance restraint was used. The distance restraint maintained an interaction between F88 and peptide 2 that was generated by Xplor-NIH and consistent with the bleaching on the PRE spectrum. Distance restraints were not applied to other sites. Whereas the 10 peptides in the Xplor-NIH models accounted for all PRE data, the four peptides were only sufficient to cover the hotspot regions. For the remaining residues on the $\alpha_1$ and $\alpha_2$ helices, four peptides only partially accounted for PRE data (*R*=0.681 for residues 29–38 and 64–74), and four peptides and was not sufficient to account for the PRE values across the molecule as a whole (*R*=0.136).

During the simulation, the nature of interactions within the hotspots changed in some cases, but the distance between peptides and bleached residues did not significantly change. The MD trajectory analysis was performed by CPPTRAJ.

## Bioinformatics analysis

Domain annotations and sequences for human proteins were obtained from the Uniprot website. Analysis was restricted to full-length, reviewed, human proteins for which there was evidence at the protein level. A Python script was used to search for consecutive Ser-Arg or Arg-Ser repeats of 4, 6, or 8 amino acids. Identification of proteins in condensates was based on databases PhaSepDB (PhaSepDB2.0 download), DrLLPs, and LLPSDB (natural protein download).

RRM domains were identified by further restricting our Uniprot search to proteins containing RRM domains of any manual assertion. In analysis of the correlation between condensation, RS repeats, and RRM domains, we found 206 proteins that together contained 365 RRM domains. Python scripts and bioinformatic data can be accessed via the github link: https://github.com/taliafargason/Repeats_in_Condensates (*Fargason, 2023*).

Percent composition values were obtained using the program LCD composer developed by Sean Cascarina in the Ross lab (*Cascarina et al., 2021*). Search was conducted for sequences 20 amino acids in length with at least 5% composition R/S. Proteins were then crossmatched against the lists of proteins containing RRM domains and proteins found in condensates. Proteins with more than one hit were only counted once (the sequence with the highest percent composition was used).

## Statistical analysis of the effect of peptide length on phase separation (*Supplementary file 1*)

Five categories of proteins were identified: proteins with no instances of the dipeptide (x=0), proteins with at least one instance of the dipeptide (x>2), proteins with at least one 4-mer dipeptide repeat (x>4), proteins with at least one 6-mer dipeptide repeat (x>6), and proteins with at least one 8-mer dipeptide repeat (x>8). Only proteins in category x=0 were automatically excluded from other categories. For instance, if a protein had a 16-mer RG repeat, it would be counted in the x>2, x>4, x>6, and x>8 categories but not x=0. Likewise, if a protein had neither 'RG' nor 'GR' anywhere in its sequence, it would be counted in the x=0 category only. Only the length of the longest uninterrupted repeat was considered. The number of repeats in the protein was not a factor in this analysis.

Within each category, the fraction of protein in condensates ($f_{ps}$)was calculated as:

$$f_{ps} = \frac{N_{ps}}{N_{ps} + N_{np}} \tag{5}$$

where $N_{ps}$ is the number of proteins with the repeat that have been found in condensates and $N_{np}$ is the number of proteins with the repeat that have not been found in condensates.

A population-based error ($E_{ps}$) was calculated as:

$$E_{ps} = \frac{1}{\sqrt{N_{ps}}} \tag{6}$$

Dipeptide repeats were considered within this error if they existed in the 8-mer form and met the criterion:

$$E_{ps(x \geq 8)} < 2 * f_{ps(x \geq 8)} \tag{7}$$

A correlation analysis was performed between the number of repeats (x) and the fraction of proteins in condensates ($f_{ps}$). Dipeptide repeats were considered to correlate significantly with phase separation if they met the criteria of: $R > 0$ (positive correlation) and $p < 0.05$.

## Statistical analysis on the effect of RS repeats and RRM domains on phase separation (Figure 1 and Figure 1—figure supplement 1)

In addition to the correlation analysis described above, Fisher's exact test and the Mann-Whitney test were used to assess the correlation between RRM domains, RS repeats, and phase separation. Because in each of these analyses, two factors were being compared against a third factor, Bonferroni's adjustment was used to set the significance threshold to $p < 0.025$. For the cases in which Fisher's exact test were used, contingency tables are included in Supplementary Files (*Supplementary files 3-4*).

Because the number of RS repeats and percent R/S composition both occur across a broad distribution, the Mann-Whitney test was employed to determine to what extent these distributions differed between proteins in condensates and proteins outside of condensates. Because RRM domains are associated with both phase separation and an increase in the number of RS repeats, these variables were separated. The Mann-Whitney test is suitable for non-normal distributions with different sample sizes (*Widen et al., 2020*). It is applicable in cases in which the sample size is greater than 30. It should be noted that the sample size of our smallest category in *Figure 1B* and *Figure 1—figure supplement 1B* (Proteins that have RRM domains but do not phase separate).

## Imaging

An SRSF1 construct (SRSF1 C16S/C148S/N220C) was tagged with Maleimide-Alexa488, dissolved into 800 mM Arg/Glu, pH 6.5, 0.2 mM TCEP, and stored at a concentration of 28 µM. Surplus Alexa488 dye was removed by a desalting column. The protein was then diluted to its final concentration in 100 mM KCl, 10 mM MES pH 6, 0.1 mM TCEP, with or without short peptides in a 96-well Cellvis glass bottom plate coated with Pluronics F127. Images are brightfield/GFP channel overlays taken on a Cytation5 imager using the software Gen5 3.10. More than three biological replicates were performed for phase separation experiments.

## Acknowledgements

We want to thank UAB Central Alabama High-Field NMR Facility. We also want to acknowledge Dr. Jinfa Ying in Ad Bax lab in at NIDDK, Dr. Charles D Schwieters at NIH for technical support. This work is supported by U.S. National Science Foundation, MCB and U.S. National Institutes of Health, NIGMS. This work was supported by the U.S National Science Foundation [MCB2024964 to JZ] and U.S National Institutes of Health [R35GM147091-01 to JZ]. Funding for open access charge: National Science Foundation and National Institutes of Health.

## Additional information

### Funding

| Funder | Grant reference number | Author |
|---|---|---|
| National Science Foundation | MCB2024964 | Jun Zhang |
| National Institutes of Health | R35GM147091 | Jun Zhang |

The funders had no role in study design, data collection and interpretation, or the decision to submit the work for publication.

### Author contributions

Talia Fargason, Data curation, Software, Formal analysis, Investigation, Visualization, Methodology, Writing - original draft, Writing - review and editing; Naiduwadura Ivon Upekala De Silva, Erin Powell, Data curation, Formal analysis; Zihan Zhang, Trenton Paul, Jamal Shariq, Steve Zaharias, Investigation; Jun Zhang, Conceptualization, Data curation, Software, Formal analysis, Supervision, Funding acquisition, Validation, Investigation, Methodology, Writing - original draft, Project administration, Writing - review and editing

### Author ORCIDs

Talia Fargason ⓘ http://orcid.org/0000-0001-6888-0356
Naiduwadura Ivon Upekala De Silva ⓘ http://orcid.org/0000-0002-5937-0271
Trenton Paul ⓘ http://orcid.org/0009-0000-0931-3888
Jun Zhang ⓘ http://orcid.org/0000-0002-5842-7424

### Decision letter and Author response

Decision letter https://doi.org/10.7554/eLife.84412.sa1
Author response https://doi.org/10.7554/eLife.84412.sa2

## Additional files

### Supplementary files

• Supplementary file 1. Effect of increasing length on phase separation for all possible dipeptide repeat combinations. Lengths (x) analyzed are x=0, x>2, x>4, x>6, and x>8. *p*-values were obtained using a two-tailed correlation analysis. A p-value of <0.05 was considered an indicator that increasing repeat length correlated significantly with fraction of proteins found in condensates. A population-based error of $\frac{1}{\sqrt{N_{ps}}}$ was used to identify whether the sample size was large enough to draw conclusions from the data (as discussed in more detail in the methods section). Whether a repeat type passed both *p*-value and population-based error criteria is indicated in the right-most column.

• Supplementary file 2. Analysis of the effect of increasing repeat length when datasets are size-matched to either n=14 (sheet 1) or n=6 (sheet 2). Percentage of proteins in condensates was found through Python's random selection tool 50 different times for each population. Average and standard deviation are indicated at the top of the sheet.

• Supplementary file 3. Contingency tables analyzing the likelihood of RRM domain and RS repeat

co-occurrence both for proteins in condensates (left) and proteins not in condensates (right). Values used correspond to those displayed in *Figure 1A*. *p*-values were obtained using Fisher's exact test.

• Supplementary file 4. Contingency tables corresponding to the *p*-values shown in *Figure 1C* and *Figure 1—figure supplement 1C*. Correlation between the fraction of proteins in condensates and the presence of a threshold number of short repeats or percentage R/S composition. *p*-values were obtained using Fisher's exact test.

• Supplementary file 5. Values corresponding to the distance between peptides and hotspot residues at various timepoints in the MD simulation. Distances (r) correspond to the length between the cysteine to which the paramagnetic center is attached and the NH hydrogen on the backbone of each residue under observation. Distances were measured using the Pymol measurement feature. Residue 42 is near to two peptides. Therefore, two distances are provided for residue 42. Bleached residues are expected to be within 12–15 Å of the paramagnetic center (*Iwahara et al., 2007*).

• MDAR checklist

### Data availability
NMR assignment has been deposited to BMRB (ID: 51299).

The following dataset was generated:

| Author(s) | Year | Dataset title | Dataset URL | Database and Identifier |
|---|---|---|---|---|
| Fargason T, Zhang J | 2022 | NMR assignment for SRSF1 | https://bmrb.io/data_library/summary/?bmrbId=51299 | Biological Magnetic Resonance Data Bank, 51299 |

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
