## [Editor Report]

This study convincingly demonstrates that the splicing factor SRSF1 can be solubilized in the presence of short RS or ER containing peptides, and uses this discovery to determine the solution NMR structure of SRSF1, as well as to map its interactions with RS peptides. These findings are important in that SR proteins are key regulators of alternative splicing but their study has been greatly hampered by their low solubility. The development of a general method that allows their structural and biochemical analysis in solution will have broad applications.

---

## [Decision Letter]

**Decision letter after peer review:**

Thank you for submitting your article "Peptides that Mimic RS repeats modulate phase separation of SRSF1, revealing a reliance on combined stacking and electrostatic interactions" for consideration by *eLife*. Your article has been reviewed by 2 peer reviewers, and the evaluation has been overseen by Douglas Black as the Reviewing Editor and James Manley as the Senior Editor. The following individual involved in the review of your submission has agreed to reveal their identity: Douglas J Kojetin (Reviewer #2).

The reviewers have discussed their reviews with one another, and the Reviewing Editor has drafted this to help you prepare a revised submission. This paper addresses a major limitation in the study of SRSF1, a critical and well-known alternative splicing factor. SRSF1 and other members of this family exhibit limited solubility due to their tendency to undergo liquid phase separation driven by their RS domains. The authors tested the hypothesis that short RS repeat peptides might compete with the intermolecular interactions that drive phase separation, and allow sufficient solubilization for biochemical and structural studies. The data convincingly show that short peptides with RS, ER, or DR repeats can render recombinant SRSF1 soluble. They go on to analyze the newly concentrated protein by NMR and present well-resolved and assignable NMR spectra of SRSF1 dissolved with RS8 as a co-solute. The authors use paramagnetic relaxation enhancement experiments to map interactions between RS8 and SRSF1. Their findings suggest that the peptide interactions with the RNA binding domains (RRMs) are similar to those of the RS domain and are mediated by a combination of ionic interactions between arginine and acidic side chains, and pi-stacking interactions with surface-exposed hydrophobic residues.

A second part of the study seeks to identify features of proteins that make them more likely to undergo phase separations. The authors use a bioinformatics approach to correlate the presence of RS repeats with the identity of proteins in databases of phase-separated material. They also use molecular modeling and software tools to predict additional RRM domains that might be prone to phase separation by looking for those containing acidic amino acids with neighboring surface-exposed aromatic/hydrophobic residues. These findings are much less convincing than the structural and biochemical studies. The findings for SR proteins, which localize to nuclear speckles are extrapolated to all kinds of RNA-binding proteins and cellular condensates. The observed fold enrichments are small. The sample sizes decrease rapidly with increasing repeat length, and it is not clear if the appropriate statistical tests were performed to correct for multiple hypothesis testing. Importantly, these predictions derived from the computational analyses are not tested experimentally.

Overall, the reviewers found the NMR and PRE data to be convincing. The use of short RS, ER, and DR peptides as co-solutes for insoluble SR-domain proteins is a novel and clever advance. The computational analyses are incomplete and over-interpreted. These findings would require experimental validation.

The reviewers and editor all found the solubilization and structural data presented in Figures 1 to 6A to be a valuable advance. However, the computational studies of Figure 6 B – F were found to be too preliminary and need validation to add much value to the manuscript. It is recommended that the authors remove Figure 6 B – F, or provide experimental validation for their interpretations of these findings. The paper should be revised to emphasize Figures 1 – 6A, to address the following essential points.

Essential revisions:

1. In Figure 1A, the sample sizes of proteins containing no RS, 2RS, 4RS etc. vary over a wide range. The authors should address whether the observed differences between percentages remain significant if size-matched subsets of the larger pools are sampled at random (multiple times). These will provide a more appropriate comparison set for the longer repeat-containing proteins. Please explain why the analysis was restricted to short RS repeats, and how short RS repeat proteins (2-8AA) can be considered representative of the SR protein family.

2. In Figure 1A, the authors should also assess other dipeptides and their repeats. It seems reasonably straightforward to repeat this analysis with every possible dipeptide to assess whether there is something special about SR, or if similar enrichments are seen with other dipeptide combinations, perhaps derived from other types of low-complexity domains.

3. In panel 1B, where multiple populations are compared against each other, have the p-values been corrected for multiple hypothesis testing? This criticism pertains to all of the informatics analyses throughout the manuscript. Additional details of how the statistical analyses were performed need to be explained in the methods.

4. For the PRE NMR-based structure calculations of the RS8/SRSF1(dRS) interactions using Xplor-NIH, additional information and analyses would improve the rigor of the reported studies. How many total independent Xplor-NIH structures were calculated? Was a single Xplor-NIH calculated structure used for MD simulations? How many independent MD simulations were performed? Was a single RS8 peptide included in the MD simulations? At what frequency are the interactions shown in Figure 5 populated in each simulation – could this be shown via distance histograms (https://amberhub.chpc.utah.edu/generating-histograms-with-cpptraj/)? Were there any restraints (PRE?) used for the MD simulations; if not, are the MD results consistent with the experimental PRE NMR data? The methods section does not describe what programs were used for MD trajectory analysis.

*Reviewer #1 (Recommendations for the authors):*

The strategy of using short RS (and similar peptides) to solubilize SFRS1 is clever, the experimental work is well done, and the NMR / PRE data are convincing. I have no major problems with this aspect of the work. The manuscript is well-written and easy to understand, with a few exceptions described below. I am less convinced by the conclusions drawn from the bioinformatic analyses. My primary concerns are detailed below.

1. In figure 1A, the sample size between no RS, 2RS, 4RS, … varies by quite a bit. Would the difference between percentages remain significant if sample size-matched subsets of the larger pools were selected at random (multiple times) for use as a comparison set with the longer repeat-containing proteins? Why was the analysis restricted to short RS repeats? How representative are short RS repeats (2-8AA) of the SR protein family?

2. In figure 1A, would you see a similar trend if you used a different dipeptide and repeats thereof? It seems reasonably straightforward to repeat this analysis with EVERY possible dipeptide to see if there is something special about SR, or if similar enrichment is seen with other dipeptide combinations.

3. In panel 1B, where multiple populations are compared against each other, have the p-values been corrected for multiple hypothesis testing? This criticism pertains to all of the informatics analyses throughout the manuscript. I looked for additional details concerning how the statistical analyses were performed in the methods, but could not find them.

4. In figure 6, it would be useful to repeat the analysis with just solved RRM structures, and just modeled RRM structures, and compare to the pooled data shown in Figure 6. Here again, it would be useful to know whether the p-values corrected for multiple hypothesis testing.

5. It should be relatively straightforward to test the model presented in the discussion, i.e. that acidic residues with neighboring hydrophobic residues are predictive of RRM proteins that localize to nuclear speckles. A mutational analysis demonstrating a change in speckle localization in cells, or even solubility in vitro, would help to test this model's validity.

*Reviewer #2 (Recommendations for the authors):*

Results section, 1st paragraph: The annotation of proteins that are present in condensates in the phase separation databases likely represents the minimal numbers as there are likely proteins not yet reported that can form or go into condensates. It would be useful to include a sentence in the Results section to indicate how this may over/underestimate the analysis in Figure 1.

Results section, 2nd subsection: Is there a published paper (that could be cited) that may have also inspired this thought? – "The high Arg composition in these proteins inspired us to use high concentrations of Arg amino acid in our protocol to purify and solubilize SRSF1." There are several published papers indicating that adding Arg to buffers can enhance general solubility.

For the PRE NMR-based structure calculations of the RS8/SRSF1(dRS) interaction using Xplor-NIH, additional information and analyses could improve the scientific rigor of the reported work. How many total independent Xplor-NIH structures were calculated? Was a single Xplor-NIH calculated structure used for MD simulations? How many independent MD simulations were performed? Was a single RS8 peptide included in the MD simulations? At what frequency are the interactions shown in Figure 5 populated in each simulation-could this be shown via distance histograms (https://amberhub.chpc.utah.edu/generating-histograms-with-cpptraj/)? Were there any restraints (PRE?) used for the MD simulations; if not, are the MD results consistent with the experimental PRE NMR data? The methods section does not describe what program(s) was(were) used for MD trajectory analysis.

Although five potential solubilizing repeat peptides were tested in Figures 2 and 3, it is possible that further enhancements to solubility could be gained from a larger structure-activity relationship (SAR) analysis. Perhaps the authors could allude to this in the manuscript (discussion?).

---

## [Author Response]

Essential revisions:1. In Figure 1A, the sample sizes of proteins containing no RS, 2RS, 4RS etc. vary over a wide range. The authors should address whether the observed differences between percentages remain significant if size-matched subsets of the larger pools are sampled at random (multiple times). These will provide a more appropriate comparison set for the longer repeat-containing proteins. Please explain why the analysis was restricted to short RS repeats, and how short RS repeat proteins (2-8AA) can be considered representative of the SR protein family.

As suggested, we did 50 random samplings using subsets of 6 and 14 proteins (Table S2). From these two sets of simulations, we observed similar percentage differences for proteins with different RS lengths. Therefore, the difference we observed is not due to sample size.

Our work is focused on RS-containing proteins, which include both SR proteins and SR-related proteins. The first SR protein, SRSF1, was discovered by Dr. Manley and Dr. Krainer over 30 years ago. The field uses their definition of the SR protein family. The SR protein family was originally defined as: any protein that has one or two N-terminal RRMs, followed by a downstream RS domain of at least 50 amino acids with > 40% RS content, characterized by consecutive RS or SR repeats. In addition to domain and amino acid composition, biological functions and posttranslational modifications are also considerations in defining SR proteins. Based on these criteria, twelve proteins are classified as SR proteins. The proteins that have RS repeats but do not meet all criteria are classified as SR-related or SR-like proteins.

It is noteworthy that the definition of SR-related proteins is ambiguous. Proposed standards are either based on percent composition(1) or number of 2mer and 4mer repeats(2), not a threshold length. Most SR or SR-like proteins have interrupted RS repeats. For this reason, the population of proteins decreases dramatically along with increased RS length. The longest uninterrupted RS region (16 aa, 8 dipeptide repeats) is found in SRSF1. Therefore, we searched for short RS repeats (2-8aa). We noticed similar trends when we searched by percent enrichment rather than repeat number, which we have added to our supplemental. A discussion of this has been added to the Results section.

2. In Figure 1A, the authors should also assess other dipeptides and their repeats. It seems reasonably straightforward to repeat this analysis with every possible dipeptide to assess whether there is something special about SR, or if similar enrichments are seen with other dipeptide combinations, perhaps derived from other types of low-complexity domains.

We appreciate this insightful comment. As suggested, we analyzed how phase separation tendency is correlated with repeat length for all dipeptide motifs. In our analysis, we assumed that the two amino acids are interchangeable. For example, RS and SR motifs are considered the same motif. In total, we analyzed 210 dipeptide motifs (190 motifs with two different amino acids, and 20 motifs with the same amino acid). Two criteria were used to select dipeptide motifs whose length is reliably correlated to phase separation. The first criterion is that the Pearson correlation p-value is smaller than 0.05. The second one is that error of the phase separation probability is smaller than half of the fraction of proteins in condensates. The error was calculated as 1/Nps, where *N_ps_* is the number of proteins with 8-mers found condensates. This estimation resembles the margin of error. The second criterion is related to the number of proteins and therefore reflects the stability of the data point. This criterion filters out the repeats that have only a few proteins (usually 1 or 2). In total, 6 motifs pass these criteria: GG, KK, QQ, PP, RG, and RS. Of these dipeptide repeats, proteins regions rich in Gly, Gln, Pro, RG/RGG, and RS have been reported to drive phase separation. It is noteworthy that 8-mer RS-containing proteins and 6-mer RS-containing proteins have the highest tendency to phase separate among all of these motifs.

3. In panel 1B, where multiple populations are compared against each other, have the p-values been corrected for multiple hypothesis testing? This criticism pertains to all of the informatics analyses throughout the manuscript. Additional details of how the statistical analyses were performed need to be explained in the methods.

We appreciate this insight. In Figure 1B, we are testing the hypothesis that the number of RS repeats is correlated with a tendency to appear in condensates. However, RRM domains are also correlated with both phase separation and RS repeats. Because many RS-containing proteins have RRM domains, we separated these two variables.

You bring up a great point that this results in two factors (RRM, phase separation) being analyzed against RS repeat number. To eliminate the possibility that random chance produces a spuriously low *p*-value, we also performed Bonferroni’s adjustment for two hypothesis testing, which would make our significance threshold 0.025 rather than 0.05. In Figure 1B, we considered the difference between columns 1 and 2 (p < 0.0001) and the difference between columns 5 and 6 (p = 0.0014) to be significant. In contrast, we found the difference between columns 3 and 4 (p = 0.139) and the difference between columns 7 and 8 (0.1449) to not be significant. These *p*-values were obtained using the Mann-Whitney test, which is suitable for a sample size > 30 as well as a situation in which sample sizes are very different and distribution is not normal. We have revised the figure legend and methods section.

4. For the PRE NMR-based structure calculations of the RS8/SRSF1(dRS) interactions using Xplor-NIH, additional information and analyses would improve the rigor of the reported studies. How many total independent Xplor-NIH structures were calculated? Was a single Xplor-NIH calculated structure used for MD simulations? How many independent MD simulations were performed? Was a single RS8 peptide included in the MD simulations? At what frequency are the interactions shown in Figure 5 populated in each simulation – could this be shown via distance histograms (https://amberhub.chpc.utah.edu/generating-histograms-with-cpptraj/)? Were there any restraints (PRE?) used for the MD simulations; if not, are the MD results consistent with the experimental PRE NMR data? The methods section does not describe what programs were used for MD trajectory analysis.

We have updated our methods section and added additional information regarding population and PRE fitting to our supplemental (Figures S7 and S8). In total, 100 Xplor-NIH structures were calculated using the ensemble size of 10. In Xplor-NIH calculations, the RRM1/RRM2 linker and RS peptides were allowed full range of motion. RRM1 was held in place, and RRM2 was allowed to move as a group. Each Xplor-NIH structure had 1 peptide (resulting in 10 peptides total in the ensemble). The ensemble structure that best fit the PRE data had a Q factor of 0.419 and Pearson Correlation Coefficient of 0.938 Using the top Xplor NIH structures, we created a starting structure for MD simulation with 4 peptides (one for each of the major PRE hotspots). This structure did not fully account for all PRE values but met the distance expectations for bleached residues (<12-15 Å). Two MD simulations are shown in this paper—one in which no restraints were applied and one in which peptide 2 was held in contact with residue F88 on the loop region following the β_4_ strand. During the simulation, the nature of interactions within the hotspots changed in some cases, but the distance between peptides and bleached residues did not significantly change. The MD trajectory analysis was performed by CPPTRAJ.